# Early Epileptic Seizures after Ischemic Stroke: Their Association with Stroke Risk Factors and Stroke Characteristics

**DOI:** 10.3390/medicina59081433

**Published:** 2023-08-08

**Authors:** Agnė Šmigelskytė, Giedrė Gelžinienė, Giedrė Jurkevičienė

**Affiliations:** Department of Neurology, Lithuanian University of Health Sciences, A. Mickevičiaus str. 9, LT-44307 Kaunas, Lithuania

**Keywords:** epileptic seizures, early seizures, ischemic stroke

## Abstract

*Background and Objectives*: A growing number of stroke survivors face various stroke complications, including new-onset epileptic seizures (ESs). Post-stroke ESs are divided into early and late ESs based on the time of onset after stroke. Early ESs are associated with worse stroke outcomes, longer hospitalization and an increased risk of late ESs. A variety of risk factors for early ESs are being studied in order to prevent their occurrence. Therefore, we aim to determine the association of early ESs with ischemic stroke risk factors and characteristics. *Materials and Methods*: A total of 166 patients, treated for ischemic stroke in the Hospital of Lithuanian University of Health Sciences Kaunas Clinics, were enrolled in a prospective cohort study. Initially, data about stroke risk factors, localization, severity and treatment were collected, followed by an observation period of 14 days for early ESs. *Results*: Early ESs occurred in 11 (6.6%) participants. The probability of early ESs after ischemic stroke among males and females (LogRank = 1.281; *p* > 0.05), younger (≤65 y) and older (>65 y) participants (LogRank = 0.129; *p* > 0.05) was the same. The presence of ischemic stroke risk factors, such as atrial fibrillation (LogRank = 0.004; *p* > 0.05), diabetes mellitus (LogRank = 1.168; *p* > 0.05) and dyslipidemia (LogRank = 0.092; *p* > 0.05), did not increase the probability of early ESs. However, participants without a prior history of arterial hypertension (LogRank = 4.453; *p* < 0.05) were more likely to develop early ESs. Stroke localization (anterior versus posterior) (LogRank = 0.011; *p* > 0.05), stroke severity (LogRank = 0.395; *p* > 0.05) and type of treatment (specific versus non-specific) (LogRank = 1.783; *p* > 0.05) did not affect the probability of early ESs.

## 1. Introduction

The current incidence of ischemic stroke worldwide is 95 cases per 100,000 inhabitants. According to the World Health Organization, the absolute number of ischemic stroke cases is rising due to rapid population growth and ageing [1]. Recent advances in stroke treatment have reduced stroke mortality. A growing number of stroke survivors face various stroke complications, including new-onset epileptic seizures (ESs) [2,3,4].

ESs occurring after stroke are divided into early and late ESs. They develop due to different pathogenetic mechanisms—biochemical changes in the ischemic brain tissue cause early ESs; however, permanent structural changes cause late ESs. There is no consensus on the time mark dividing early and late ESs—in various studies, it ranges from 24 h to 1 month, but usually it is 1 or 2 weeks [4]. This causes a high variability in the reported incidence, risk factors, clinical presentation and treatment options of early and late ESs among different studies and makes it difficult to draw accurate conclusions.

Early ESs are associated with an increased risk of late ESs [5,6] and worse stroke outcomes [7,8]. Furthermore, some early ESs may be missed due to a non-convulsive manifestation, leading to inadequate treatment, prolonged hospitalization and an extensive neuronal injury [9,10,11].

Various stroke risk factors have been associated with an increased risk of early ESs. Most studies link atrial fibrillation with early ESs [12,13] as cardioembolic strokes often lead to cortical damage [14]. The findings about other stroke risk factors have been inconsistent. One study has shown that arterial hypertension on admission has a protective effect against early ESs after ischemic stroke as higher blood pressure leads to a better brain perfusion and a smaller ischemic area [15]. On the other hand, high blood pressure increases the risk of the hemorrhagic transformation of ischemic strokes [16], and brain hemorrhage is a well-known risk factor of early ESs [17]. In some studies, hyperglycemia due to diabetes or other reasons was linked with an increased risk of early ESs [18]. Dyslipidemia is rarely mentioned as an independent risk factor. However, alongside arterial hypertensions and diabetes, it contributes to the development of small cerebral artery disease [18], which, according to some studies, increases the risk of early ESs [19]. Determining stroke risk factors that increase the risk of early ESs is important as their tighter control would not only lead to a lower incidence of stroke but also early ESs.

Some studies found temporal or frontal lobe strokes to carry a higher risk of early ESs [20], while others found this association in strokes of the posterior cerebral artery territory [21]. Stroke severity (NIHSS on admission) has also been named as an ES risk factor [21]. Stroke reperfusion therapy has also been linked to a higher incidence of ESs; however, recent studies have failed to find an association [6,22,23]. This is an important issue to examine as reperfusion therapies are becoming more accessible to a large numbers of patients worldwide [24,25,26].

Therefore, the aim of our study is to determine the association of early ESs with ischemic stroke risk factors and stroke characteristics.

## 2. Materials and Methods

### 2.1. Study Design

We conducted a prospective cohort study in the Hospital of Lithuanian University of Health Sciences Kaunas Clinics Department of Neurology from 1 February 2018 to 31 January 2020.

Data about age, gender, stroke risk factors, localization, severity and treatment were collected from medical documents. Patients were observed in the stroke unit or later in the rehabilitation center for the first 14 days after acute ischemic stroke for the occurrence of early ESs.

This study was approved by the Bioethics center of the Hospital of Lithuanian University of Health Sciences Kaunas Clinics (BEC-MF-208).

### 2.2. Selection of Participants

We enrolled adult (≥18 years) patients with an initial diagnosis of acute ischemic stroke. Patients with head trauma, intracerebral hemorrhage, subarachnoid hemorrhage, epidural hemorrhage, primary or secondary tumors of the central nervous system and diagnosis of epilepsy prior to stroke were not included in this study.

We asked 243 patients with an initial acute ischemic stroke diagnosis or their relatives (in the case of aphasia and coma) to participate in our study—193 patients agreed. The final acute ischemic stroke diagnosis was made based on clinical neurological signs and deficits, computer tomography imaging and magnetic resonance imaging in certain cases. After a full neurological workup, 8 patients were diagnosed with other conditions than acute ischemic stroke (such as hemiplegic migraine). Seventeen patients developed hemorrhagic transformations of their ischemic strokes. Since brain hemorrhages are likely to increase the risk of ESs and we considered 17 patients to be an insufficient amount to form a separate group (only one of them developed early ESs), they were excluded. During the observation period, 2 people had a recurrent stroke; therefore, they were also excluded from the study. The finale sample size consisted of 166 acute ischemic stroke patients.

### 2.3. Gathered Data

Clinical data about age, gender, stroke risk factors (arterial hypertension, diabetes mellitus, atrial fibrillation and dyslipidemia), stroke localization, severity, occurrence of early ESs and time from stroke to ESs were collected.

-The age of the study participants varied between 38 and 94 years. According to age, they were divided into younger (≤65 years) and older (>65 years) participants.-According to gender, the participants were divided into male and female.-Concerning stroke risk factors, the participants were divided into groups with or without arterial hypertension, with or without atrial fibrillation, with or without diabetes mellitus and with or without dyslipidemia (out of 166 patients, 121 were evaluated for dyslipidemia).-According to stroke localization, the participants were divided into groups with anterior and posterior circulation strokes. We considered dividing participants into smaller groups (such as affected lobes and affected blood vessels); however, due to a relatively small sample size, we were not able to.-Ischemic stroke severity was determined using the National Institutes of Health Stroke Scale (NIHSS). NIHSS was assessed for 87 out of the 166 participants. Based on stroke severity, the participants were divided into 3 groups: NIHSS ≤ 8, NIHSS > 8 < 16 and NIHSS ≥ 16.

Early ESs were considered ESs that occurred during the first 2 weeks (≤14 days) after ischemic stroke.

### 2.4. Data Analysis

The analysis of the data was performed using SPSS 23 software program. Sample size normality was assessed using a Shapiro–Wilk test. Data were analyzed using a Mann–Whitney U test, χ^2^ test and Kaplan–Meier analysis. Statistical significance was assumed when *p* < 0.05.

## 3. Results

### 3.1. Demographic and Clinical Characteristics of the Study Participants

From a total of 166 participants, 94 (56.6%) were male and 72 (43.4%) were female. The mean age of the participants during stroke was 68.1 ± 11.7 years. The females were older than the males (*p* < 0.05). Other demographic and clinical characteristics of the study participants are presented in Table 1.

Early ESs occurred in 11 (6.6%) participants. The majority of early ES cases (seven (63.6%)) occurred during the first day, two (18.2%) ES cases occurred during the second day, one (9.1%) ES case occurred during the third day and one (9.1%) ES case occurred during the fourth day after ischemic stroke. All ES cases occurred within the first four days after stoke (Figure 1).

### 3.2. The Association of Ischemic Stroke Risk Factors with the Occurrence of Early Epileptic Seizures

The probability of early ESs after ischemic stroke among males (*n* = 94) and females (*n* = 72) (LogRank = 1.281; *p* > 0.05), younger (≤65 y) and older (>65 y) participants (LogRank = 0.129; *p* > 0.05) was the same (Figure 2).

Participants without a prior history of arterial hypertension (*n* = 35) had a higher probability (LogRank = 4.453; *p* < 0.05) of developing early ESs after ischemic stroke than participants with a history of arterial hypertension (*n* = 131). The probability of early ESs after ischemic stroke among participants with and without atrial fibrillation (LogRank = 0.004; *p* > 0.05), with and without diabetes mellitus (LogRank = 1.168; *p* > 0.05) and with and without dyslipidemia (LogRank = 0.092; *p* > 0.05) was the same (Figure 3).

### 3.3. The Association of Ischemic Stroke Characteristics with the Occurrence of Early Epileptic Seizures

The probability of early ESs among participants with anterior (*n* = 119) and posterior (*n* = 47) brain circulation ischemic strokes (LogRank = 0.011; *p* > 0.05), ischemic strokes of different severities (NIHSS ≤ 8 (*n* = 57), NIHSS > 8 < 16 (*n* = 23) and NIHSS ≥ 16 (*n* = 7)) (LogRank = 0.395; *p* > 0.05) and specific (*n* = 93) and non-specific (*n* = 73) ischemic stroke treatments (LogRank = 1.783; *p* > 0.05) was the same (Figure 4).

Due to the small number of early ES cases, we did not divide our study participants into other smaller groups for further statistical analysis.

## 4. Discussion

According to our findings, early ESs after ischemic stroke occurred in 6.6% of participants. Other studies using 2 weeks as the cut-off point for early ESs reported a slightly lower incidence—0.5–5.4%. Since all ES cases in our study occurred during the first four days after ischemic stroke, our findings are comparable with studies using 1 week as the cut-off point. They reported an incidence in the range of 1–13.6% [4,27], which is in line with our data.

We found that the majority of early ES cases (seven (63.6%)) occurred during the first day after ischemic stroke. Other studies described a similar distribution in the time of early ESs. According to Feher et al., 73.6% of ES cases occurred during the first day after ischemic stroke [28]. In the work of Brigo et al., the median time from ischemic stroke to early ESs was 2 days and 75.9% of acute ESs occurred during the first 3 days [29]. However, both Feher et al. and Brigo et al. did not exclude patients with hemorrhagic transformations from their studies.

Most studies did not associate certain age or gender with an increased incidence of early ESs [6]. However, Zöllner et al. conducted a population-based study in 2020 involving around 140 000 cases and identified older age and female gender as risk factors [30]. Our data are not consistent with those of Zöllner et al., but our male participants were younger than the females ones. This might influence our results concerning age and gender as risks factors for early ESs.

A systematic review and meta-analysis conducted by Ma et al. concluded that arterial hypertension is not an independent risk factor for early ESs in the case of stroke [17,31]. However, there is some scientific evidence that arterial hypertension may protect from developing early ESs. Our study participants with a prior history of arterial hypertension had a lower risk of developing early ESs. In a 2016 study by Hundozi et al., patients who arrived at the Emergency Room with ischemic stroke and a low or normal blood pressure were found to have a 2.5 times higher risk of developing early ESs. The authors hypothesized that a low or normal blood pressure is responsible for the formation of a larger ischemic area, which leads to an increased risk of ESs [15]. Indeed, other large-scale studies showed that intensive blood pressure, lowering in acute ischemic stroke, leads to cerebral hypoperfusion, larger ischemic core, poorer functional outcome and increased mortality [32,33,34].

It is important to note that chronic arterial hypertension and an acute elevation of blood pressure in the case of cerebral vascular occlusion might not bear the same protective effect. Moreover, a high blood pressure might result in the hemorrhagic transformation of an ischemic stroke, leading to an increased risk of ESs [16,17,35]. Therefore, further studies about the possible neuroprotective effect of higher blood pressure against early ESs are required.

Scientific data about the association of stroke localization and ES remain contradictory. According to our study, the probability of early ESs among participants with anterior and posterior brain circulation ischemic strokes was the same. Yamada et al. described frontal lobe strokes to carry a higher risk of ESs [36]. Ouerdiene et al. associated middle cerebral artery strokes with a higher incidence of early ESs [37], while Ferreira-Atuesta et al. found this association with posterior cerebral artery strokes [6]. Chou et al. used a quantitative analysis of ischemic lesions in brain magnetic resonance imaging to identify hot spots for early ESs. They found the majority of hot spots for early ESs to be located near the left central region, superior parietal lobule, lateral temporal cortex and right medial occipital cortex [38]. However, the majority of studies indicated that lesions of the cerebral cortex, not the subcortex, regardless of their location, increase the risk of early ESs [36,37,38].

It was hypothesized that stroke reperfusion treatment may directly or indirectly cause ESs [13]. The fact that reperfusion treatment is more common in cases of severe stroke (which is a risk factor for ESs) might have led to the impression of causation. Nevertheless, recent scientific data gravitate towards the conclusion that the recombinant tissue plasminogen activator used for thrombolysis probably does not have a direct pro-epileptic effect or the effect is insignificant [6,39,40,41]. The reperfusion injury of ischemic tissue after successful recanalization could indirectly contribute to the development of early ESs. However, large trails [6,22,23] and systemic reviews [42] did not associate reperfusion treatment, except in cases of hemorrhagic transformation, with an increased risk of early ESs.

Our study had several limitations. The main drawback of our study is the small sample size. A study with a larger number of participants should be performed to obtain more accurate results. Additionally, gathering a sufficient group of patients with hemorrhagic transformation would be ideal. Furthermore, there is a statistically significant age difference between our male and female participants. Although females experience acute ischemic stroke when they are approximately 4–6 years older than males [43,44], groups of different sexes in studies should not differ in age to obtain accurate results. On the other hand, our sample size might better reflect the situation of the real world. Finally, some patients might have experienced non-convulsive ESs that we were not aware of as we do not routinely use EEG monitoring for our stroke patients.

## 5. Conclusions

In our study group, early ESs occurred in 6.6% of the participants. The majority of early ES cases (63.6%) occurred during the first day after stroke, and all ESs occurred within the first four days after stroke.

The probability of early ESs was not affected by age, gender, stroke risk factors (atrial fibrillation, diabetes mellitus and dyslipidemia), stroke severity, location and type of treatment. The only clinical finding increasing the probability of early ESs was the lack of a prior history of arterial hypertension.

## Figures and Tables

**Figure 1 medicina-59-01433-f001:**
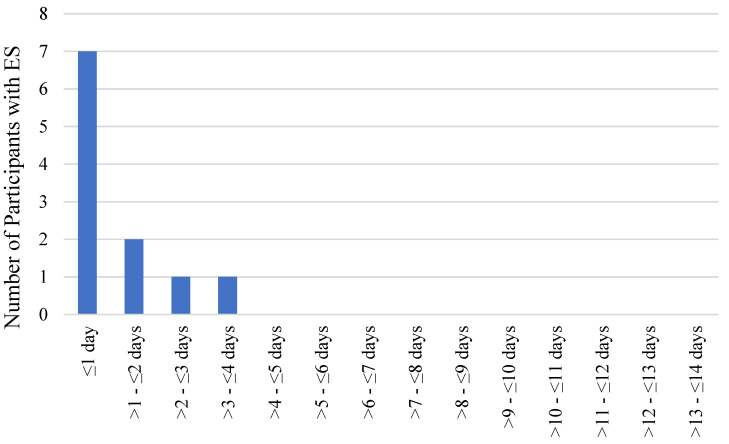
The occurrence of early ESs during the first 14 days after ischemic stroke.

**Figure 2 medicina-59-01433-f002:**
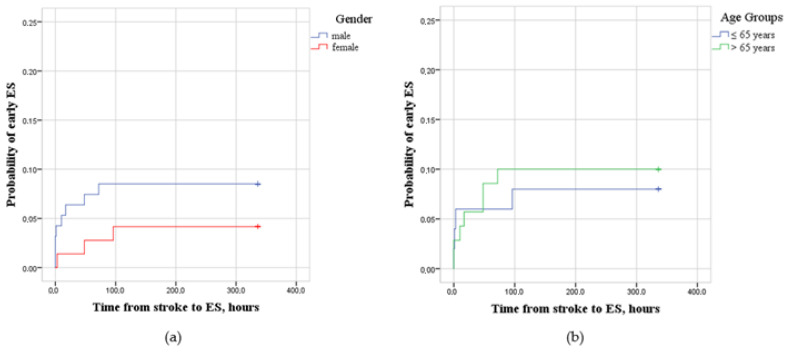
(**a**) Probability of early epileptic seizures (ESs) after ischemic stroke for male (*n* = 94) and female (*n* = 72) participants. (**b**) Probability of early epileptic seizures (ESs) after ischemic stroke for younger (≤65 years) (*n* = 70) and older (>65 years) (*n* = 96) participants.

**Figure 3 medicina-59-01433-f003:**
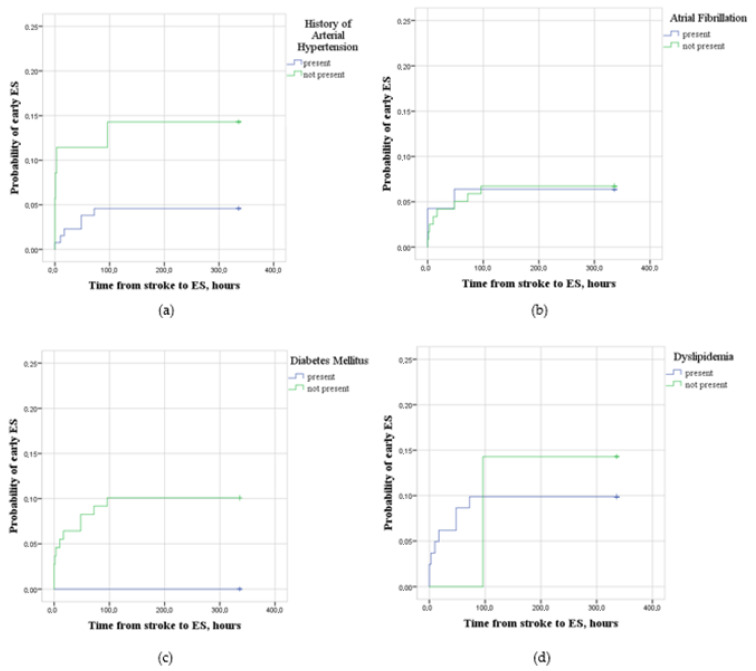
(**a**) Probability of early epileptic seizures (ESs) after ischemic stroke for participants with (*n* = 131) and without (*n* = 35) a prior history of arterial hypertension. (**b**) Probability of early epileptic seizures (ESs) after ischemic stroke for participants with (*n* = 47) and without (*n* = 119) atrial fibrillation. (**c**) Probability of early epileptic seizures (ESs) after ischemic stroke for participants with (*n* = 23) and without (*n* = 143) diabetes mellitus. (**d**) Probability of early epileptic seizures (ESs) after ischemic stroke for participants with (*n* = 113) and without (*n* = 8) dyslipidemia.

**Figure 4 medicina-59-01433-f004:**
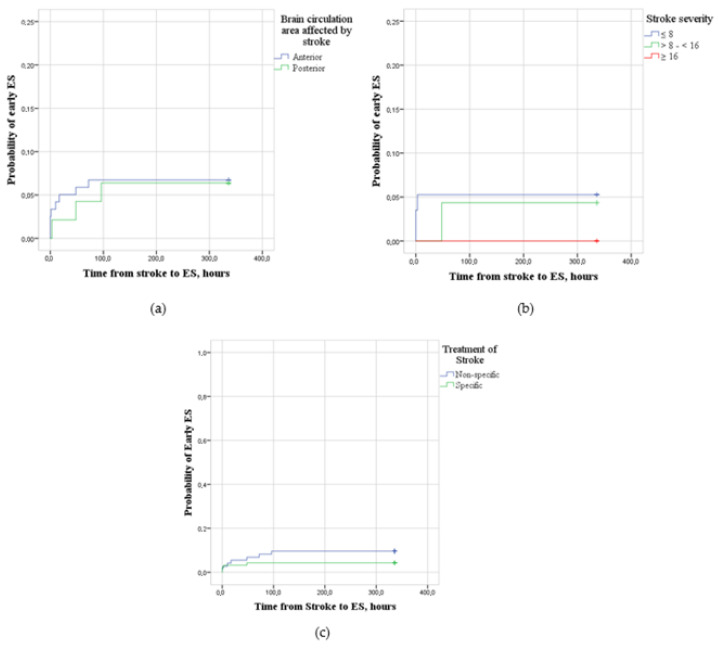
(**a**) Probability of early epileptic seizures (ESs) after ischemic stroke for participants with anterior (*n* = 119) and posterior (*n* = 47) brain circulation strokes. (**b**) Probability of early epileptic seizures (ESs) after stroke of different severities—NIHSS ≤ 8 (*n* = 57), NIHSS > 8 < 16 (*n* = 23) and NIHSS ≥ 16 (*n* = 7). (**c**) Probability of early epileptic seizures (ESs) after non-specific (*n* = 73) and specific (*n* = 93) ischemic stroke treatment.

**Table 1 medicina-59-01433-t001:** Demographic and clinical characteristics of the study participants.

Demographic and Clinical Characteristics	Study Participants (*n* = 166)
**Gender**	
Male, *n* (%)	94 (56.6)
Female, *n* (%)	72 (43.4)
**Age** (years), m ± SD	68.1 ± 11.7
Males	66.1 ± 11.0 *
Females	70.8 ± 12.1
**Age Groups**	
≤ 65 years, *n* (%)	70 (42.2)
> 65 years, *n* (%)	96 (57.8)
**Stroke Risk Factors**	
Arterial hypertension, *n* (%)	131 (78.9)
Atrial fibrillation, *n* (%)	47 (28.3)
Diabetes mellitus, *n* (%)	23 (13.9)
Dyslipidemia (*n* = 121), *n* (%)	113 (93.4)
**Brain Circulation Affected by Stroke**	
Anterior, *n* (%)	119 (71.7)
Posterior, *n* (%)	47 (28.3)
**Stroke Severity (*n* = 87)**	
NIHSS ≤ 8, *n* (%)	57 (65.5)
NIHSS > 8 < 16, *n* (%)	23 (26.4)
NIHSS ≥ 16, *n* (%)	7 (8.0)
**Treatment of Stroke**	
Non-specific, *n* (%)	73 (44.0)
Specific, *n* (%)	93 (56.0)
**Specific Treatment of Stroke**	
Thrombolysis, *n* (%)	77 (46.4)
Mechanical thrombectomy, *n* (%)	3 (1.8)
Thrombolysis and mechanical thrombectomy, *n* (%)	13 (7.8)

* *p* < 0.05, Mann–Whitney U test—compared to females; ES—epileptic seizure; NIHSS—National Institutes of Health Stroke Scale.

## Data Availability

Data sharing is not applicable to this article.

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
