# Peer review of "Early Epileptic Seizures after Ischemic Stroke: Their Association with Stroke Risk Factors and Stroke Characteristics"

_medicina, 2023, doi:10.3390/medicina59081433_

Round 1
Reviewer 1 Report
In Šmigelskytė et al., the authors provide a clinical perspective to show the association of early epileptic seizures after ischemic stroke. The authors state that in recent times, while the stroke mortality has reduced due to recent advancements, early onset epileptic seizures (ES) have emerged as anew complications in stroke survivors. The authors show that in their study group of 166 patients, ES occurred in 6.6% of participants. The majority of early ES (63.6%) occurred during the first day after stroke, all ES occurred within the first four days after stoke. ES occurred in 6.6% of participants. The majority of early ES (63.6%) occurred during the first day after stroke, all ES occurred within the first four days after stoke. The study is well designed taking into consideration all ages and sex, and the statistics used are good. This study is of some importance in the field of ischemic stroke. However, the authors should cite more recent references to support their findings.
Author Response
Dear Reviewer,
thank You for Your comments and insights. I have revised my article accordingly - cited more recent references, described methods in more detail and added commentary to the discussion section.
Kind regards,
A. Šmigelskytė
Reviewer 2 Report
This is a good study, but which could have been improved to arrive to some other important conclusions. The overreaching disadvantage of this study is the small population size that was recruited. I agree with the authors that the low number of male participants over female might have been a factor that effected the results from this study concerning age and gender.
As to the hypothesis that normal or low blood pressure might be responsible for the formation of a larger lesion that increases ES makes sense. It can also be speculated that the higher blood pressure might be one factor that prevents ES as the higher pressure pushes the cerebral perfusion pressure and prevents hypoperfusion and is therefore neuroprotective.
Reperfusion injury can only be considered in the context of ES after stroke when there is spontaneous or forced recanalization after an extended period of hypoperfusion within ischemic limits.
As to stroke location and its influence on early ES, I tend to push foward the idea that this all has to do with the extent of collateralization and which as we know is highly variable in humans and the depth (or degree of blood flow reduction, coupled to its duration) of the ischemic episode.
It would have been of interest to study early ES on those indivduals who have suffered a secondary stroke, including the age and gender differentiation.
In my opinion, I would accept this review as a good initiative of a small study with its own limitations and if the above points are included and expanded in the conclusion section.
Author Response
Dear Reviewer,
thank You for Your comments. I have greatly improved the discussion section according to Your insights.
I have presented various nuances of the possible neuroprotective effect of arterial hypertension in more detail.
I have edited the paragraph about reperfusion treatment, reperfusion injury and the risk of early ES. I hope I have presented our findings and relevant literature in a clearer manner.
As to stroke location and its influence on early ES, there are no scientific studies about the association of stroke location, collateralization and early ES. It could be one of the confounding factors, but it is very hard to prove. As you mentioned, it is highly variable in individuals and thus difficult to study. I have chosen not to mention collateralization in the discussion. However, I have improved this discussion paragraph to better reflect the current understanding of the influence of stroke location on the development of early ES.
Indeed, it would have been of interest to study early ES in those individuals who have suffered a secondary stroke. However, we did not have enough patients with secondary stroke to form a separate group of participants.
Kind regards,
A. Šmigelskytė